# Association between oral frailty and cystatin C-related indices—A questionnaire (OFI-8) study in general internal medicine practice

**Hiroshi Kusunoki**[1,2,3]*, **Kazumi Ekawa**[1,4], **Nozomi Kato**[1], **Keita Yamasaki**[1,5], **Masaharu Motone**[1,6], **Ken Shinmura**[2], **Fumiki Yoshihara**[3], **Hideo Shimizu**[1]

**1** Department of Internal Medicine, Osaka Dental University, Hirakata, Osaka, Japan, **2** Department of General Internal Medicine, Hyogo Medical University, Nishinomiya, Hyogo, Japan, **3** Department of Hypertension and Nephrology, National Cerebral and Cardiovascular Center, Suita, Osaka, Japan, **4** Department of Environmental and Preventive Medicine, Hyogo Medical University, Nishinomiya, Hyogo, Japan, **5** Department of Health and Sports Sciences Graduate School of Medicine, Osaka University, Suita, Osaka, Japan, **6** Faculty of Health Sciences, Osaka Dental University, Hirakata, Osaka, Japan

* kusunoki1019@yahoo.co.jp

**Data Availability Statement:** Data cannot be released because it contains personal patient information. Researchers who meet the criteria for

## Abstract

### Background

Cystatin C-related indices such as the ratio of creatinine to cystatin C (Cr/CysC) and the ratio of estimated glomerular filtration rate by cystatin C (eGFRcys) to creatinine eGFRcre (eGFRcys/eGFRcre) levels have been shown to be associated with muscle mass and strength and can be markers of sarcopenia. Oral frailty is defined as an age-related gradual loss of oral functions, accompanied by a decline in cognitive and physical functions. It results in adverse health-related outcomes in older age, including mortality, physical frailty, functional disability, poor quality of life, and increased hospitalization and falls. Therefore, poor oral health among the elderly is an important health concern due to its association with the pathogenesis of systemic frailty, suggesting it to be a multidimensional geriatric syndrome. The Oral Frailty Index-8 (OFI-8) is a questionnaire that can be used for easy screening of oral frailty. This study aimed to investigate whether cystatin C- related indices are different between patients with low to moderate risk of oral frailty and those at high risk of oral frailty, using the OFI-8 in attending a general internal medicine outpatient clinic.

### Materials and methods

This is a cross-sectional study that included 251 patients with a mean age of 77.7±6.6 years and a median age of 77 years (128 men: mean age, 77.1±7.3 years; median age, 77 years and 123 women: mean age, 78.4±5.7 years; median age, 78 years) attending general internal medicine outpatient clinics. OFI-8 scores were tabulated by gender to determine whether there were differences between patients at low to moderate risk of oral frailty (OFI-8 score ≤3 points) and those at high risk (OFI-8 score ≥4 points) in Cr/CysC, eGFRcys/eGFRcre levels, height, weight, grip strength, etc. were examined.

access to confidential data may obtain the data via the Clinical Trials Committee. The name of the institution limiting this data is the IRB of Osaka Dental University. If you would like to access the data, please contact us by phone (+81-72-864-3111) or online (https://www.osaka-dent.ac.jp/access.html).

**Funding:** This study was supported in part by JSPS KAKENHI (grant number: 19K16995 [2019-2022]), awarded to HK.

**Competing interests:** The authors have declared that no competing interests exist.

## Results

The OFI-8 score was higher in women than in men, suggesting that oral frailty is more common in women. Cr/CysC, eGFRcys/eGFRcre and grip strength were significantly lower in both men and women in the high-risk group for oral frailty (OFI-8 score $\geq$ 4). Height, hemoglobin level, red blood cell count, and serum albumin levels were significantly lower in men with an OFI-8 score $\geq$4. Receiver operating characteristic curve (ROC) analysis also showed that Cr/CysC and eGFRcys/eGFRcre were significantly associated with an OFI-8 score$\geq$4 in both men and women.

## Conclusion

Cr/CysC and eGFRcys/eGFRcre were significantly lower in the high-risk group for oral frailty on the OFI-8in both men and women. A relationship exists among cystatin C-related indices, which can effectively screen systemic frailty. Similarly, the OFI-8 score can be used to effectively screen oral frailty. Thus, a collaboration that incorporates both systemic and oral frailty from medical and dental perspectives is required.

## Introduction

Oral health is an essential aspect of health, life satisfaction, QoL, and self-perception. Impairment of oral function is highly common in older adults, and aging has been reported to indirectly interact with several frailty domains through multiple pathways. An overt example of this relationship is age-related functional oral deterioration, characterized by poor dental hygiene, inadequate dental prostheses, and dietary deficiencies, resulting in a high risk of nutritional frailty [1].

Oral frailty is defined as an age-related gradual loss of oral function, accompanied by a decline in cognitive and physical functions. It results in major adverse health-related outcomes in older age, including mortality, physical frailty, functional disability, loss of quality of life, and reduced hospitalization and falls [2].

Poor oral health among the elderly is an important health concern due to its association with the pathogenesis of systemic frailty, suggesting it to be a multidimensional geriatric syndrome. Therefore, oral frailty may be a risk factor for systemic frailty [3].

Traditionally, oral frailty, physical frailty, and sarcopenia have been treated individually from the medical and dental perspectives. In the past, the medical side has focused on elderly patients with advanced systemic frailty, such as those with progressive disease, such as heart failure and respiratory dysfunction, falls, and bone fractures, which may lead to the development of musculoskeletal disorders. By approaching and evaluating oral frailty from a medical perspective, a possibility exists for early detection of potential frailty, sarcopenia, and therapeutic intervention.

When oral frailty progresses to more apparent oral hypofunction, the Japan Dental Association recommends dentists to evaluate seven items: 1) oral hygiene, 2) oral dryness, 3) occlusal force, 4) tongue-lip motor function, 5) tongue pressure, 6) masticatory function, and 7) swallowing function [4]. However, these measurements require specialized training and specialized precision equipment in the dental office.

In addition to clinical examinations, questionnaires have been developed for screening for oral frailty. Tanaka et al. previously proposed an eight-item questionnaire, the Oral Frailty

Index-8 (OFI-8), to facilitate the screening of older adults at risk for oral frailty in a community setting. The OFI-8 integrates oral health-related behaviors and associated indicators of oral frailty [5].

However, if an index that reflects oral function can be established using blood test data and physical measurement results obtained in daily medical care, it would be possible to easily detect patients at a high risk of oral frailty in medical practice and provide early dental intervention, thereby promoting medical-dental collaboration. We have previously reported the relationship between cystatin C (CysC), an indicator of renal function that, unlike creatinine (Cr), is not easily affected by muscle mass and sarcopenia in an epidemiological study at Hyogo Medical University [6, 7].

Several reports from Japan and abroad indicate that the ratio of creatinine to cystatin C (Cr/CysC) reflects muscle mass [8–20]. We and other groups have reported that the ratio of estimated glomerular filtration rate by cystatin C (eGFRcys) to creatinine eGFRcre (eGFRcys/eGFRcre) reflects muscle mass [6, 21]. eGFRcys is significantly associated with frailty and sarcopenia but not eGFRcre [22–24]. We have additionally focused on the association between oral function and cystatin C and recently reported that eGFRcys and eGFRcys/eGFRcre are associated with low tongue pressure [25].

Therefore, in this study, we aimed to examine whether cystatin C-related indices (Cr/CysC and eGFRcys/eGFRcre), which have been shown to be significantly associated with muscle mass and muscle strength, are associated with decreased oral function assessed using the OFI-8. Additionally, we examined the association between oral dysfunction assessed using the OFI-8 and body size, grip strength, and other blood test indices.

## Materials and methods

This was a cross-sectional study. All patients aged $\geq 65$ years enrolled in this study were Japanese and provided informed consent to participate. We enrolled 251 patients with a mean age of 77.7±6.6 years and a median age of 77 years (128 men: mean age, 77.1±7.3 years; median age, 77 years and 123 women: mean age, 78.4±5.7 years; median age, 78 years) who were admitted to our institutions (Osaka Dental University and the National Cerebral and Cardiovascular Center) between April 2022 and December 2022.

In this study, blood tests including Cr and CysC, physical measurements (height and weight), grip strength measurements, and the OFI-8 questionnaire were conducted on a total of 251 outpatients at the Department of Internal Medicine, Osaka Dental University Hospital and the Department of Nephrology and Hypertension, National Cerebral and Cardiovascular Center. Oral assessments using the questionnaire were performed by five internal medicine outpatient physicians.

The results of the questionnaire were tabulated, and the participants were divided into two groups by sex: a low- to moderate-risk group for oral frailty (OFI-8 score≤3) and a high-risk group for oral frailty (OFI-8 score≥4). We analyzed the differences in blood data, physical parameters, and grip strength between these groups.

We measured the maximum grip strength using a grip strength tester (GRIP-A; Takei Ltd., Niigata, Japan) [26]. We calculated eGFRcre and eGFRcys using equations from the Japanese Society of Nephrology [27, 28]. Receiver operating characteristic curve (ROC) analysis was performed to confirm the diagnostic efficacy of Cr/CysC and eGFRcys/eGFRcre for OFI $\geq 4$, and the area under the curve (AUC) was calculated.

The study protocol was approved by the ethics committees of our institutions (Osaka Dental University and National Cerebral and Cardiovascular Center). All procedures performed in studies involving human participants were in accordance with the ethical standards of the

institutional and/or national research committee where the studies were conducted (IRB approval number 2022–25 at Osaka Dental University) and with the 1964 Helsinki Declaration and its later amendments or comparable ethical standards. All of the patients enrolled in this study were Japanese and gave informed consent to participate in this study.

## Questionnaire

The Oral Frailty Index-8 (OFI-8) was used in this study. The questionnaire consists of eight items and is widely used in Japan.

1. Do you have any difficulties eating tough foods compared 6 months ago? (Yes)

2. Have you choked on your tea or soup recently? (Yes)

3. Denture use (Yes)

4. Do you often experience having a dry mouth? (Yes)

5. Do you go out less frequently than you did last year? (Yes)

6. Can you eat hard foods like squid jerky or pickled radish? (No)

7. How many times do you brush your teeth per day? (<3 times/day)

8. Have you visit dental clinic at least annually? (No)

   Using the standard protocol, if subjects answered "yes" to Items 1, 2, or 3, two points were given for each answer. If the subjects answered "yes" to Items 4 and 5, one point was given for each answer. If the subjects answered "no" to Items 6, 7, or 8, one point was given for each answer. The maximum possible score was 11. The screening criterion was defined as the sum of the scores called OFI-8 scores. The higher the OFI-8 score, the higher the risk of oral frailty, that is, 0–2 points indicated low risk; 3 points, moderate risk; and greater than 4 points, high risk. In this study, all subjects were divided by sex, OFI-8 score ≤3 (low-to moderate-risk group), and OFI-8 score ≥4 (high-risk group), and the characteristics of each group were examined.

## Statistical analysis

The results are expressed as the mean±standard deviation (SD) or percentage. For intergroup comparisons, Student's t-test was used. Categorical variables are expressed as absolute (n) and relative frequency (%), and they were analyzed using Fisher's exact test. ROC analysis was performed to confirm the diagnostic efficacy of Cr/CysC and eGFRcys/eGFRcre for OFI ≥4. The area under the curve (AUC) was calculated. The JMP 13.1 software was used for data analysis. Statistical significance was set at p <0.05.

## Results

The baseline characteristics, indices of physical examination, blood examination results, and comorbidity frequencies of the participants are presented in Table 1. The mean age of both men and women was approximately 78 years. Body size and grip strength were naturally larger in men; thus, they showed higher Cr, reflecting muscle mass, and higher Cr/CysC, since CysC showed no difference with respect to sex. eGFRcys and eGFRcys/eGFRcre were also higher in men. Red blood cell count (RBC), hemoglobin (Hb), and hematocrit (Ht) were higher in men than in women, suggesting that women tend to be anemic. Additionally, complications such as dyslipidemia and osteoporosis tended to be higher in women.

**Table 1. Baseline characteristics, physical examination indices, blood examination results, and comorbidity frequency of the participants.**

| | Total (n = 251) | Men (n = 128) | Women (n = 123) | p |
|---|---|---|---|---|
| Age (years) | 77.7±6.6 | 77.1±7.3 | 78.4±5.7 | 0.131 |
| Height (cm) | 158.2±9.5 | 164.9±7.1 | 151.2±6.0 | <0.001 |
| Weight (kg) | 58.8±11.5 | 65.2±9.5 | 52.0±9.2 | <0.001 |
| Body mass index: BMI | 23.4±3.4 | 23.9±2.6 | 22.8±4.0 | 0.006 |
| Grip strength (kg) | 23.2±9.2 | 29.6±8.0 | 16.5±4.5 | <0.001 |
| Cr (mg/dL) | 0.95±0.35 | 1.06±0.34 | 0.84±0.32 | <0.001 |
| CysC (g/dL) | 1.26±0.42 | 1.26±0.39 | 1.26±0.45 | 0.987 |
| Cr/CysC | 0.77±0.17 | 0.86±0.15 | 0.68±0.13 | <0.001 |
| eGFRcre (mL/min/1.73 m$^2$) | 56.3±16.0 | 57.3±16.3 | 55.3±15.6 | 0.320 |
| eGFRcys (mL/min/1.73 m$^2$) | 55.8±18.0 | 57.6±17.6 | 53.9±18.3 | 0.103 |
| eGFRcys/eGFRcre | 1.00±0.22 | 1.02±0.22 | 0.98±0.22 | 0.171 |
| Red blood cells (×10$^4$/μL) | 423±52 | 435±52 | 410±50 | <0.001 |
| Hemoglobin (g/dL) | 13.2±1.5 | 13.8±1.4 | 12.7±1.4 | <0.001 |
| Hematocrit (%) | 39.6±4.3 | 41.0±4.1 | 38.3±4.1 | <0.001 |
| Total protein (g/dL) | 7.1±0.5 | 7.1±0.5 | 7.2±0.5 | 0.094 |
| Albumin (g/dL) | 4.2±0.3 | 4.2±0.3 | 4.2±0.3 | 0.561 |
| Hypertension, n(%) | 225(89.6) | 116(90.6) | 109(88.6) | 0.681 |
| Diabetes, n(%) | 50(19.9) | 29(22.7) | 21(17.1) | 0.343 |
| Dyslipidemia, n(%) | 141(56.2) | 64(50.0) | 77(62.6) | 0.056 |
| Osteoporosis, n(%) | 27(10.8) | 3(2.3) | 24(19.5) | <0.001 |
| Malignant neoplasm, n(%) | 8(3.2) | 5(3.9) | 3(2.4) | 0.723 |
| Cardiovascular disease, n(%) | 62(24.7) | 34(26.6) | 28(22.8) | 0.559 |
| Cerebrovascular disease, n(%) | 24(9.6) | 12(9.4) | 12(9.8) | 1.000 |

Data are expressed as mean ± SD.

The results of the OFI-8 are presented in Table 2. The overall score was significantly higher for women, with low- to moderate-risk scores of 3 or less tending to be higher in men and high-risk scores of 4 or more tending to be higher in women, suggesting that oral frailty is more common in women. Looking at each question, women were more likely to have difficulties eating tough foods compared to 6 months ago, choke on tea or soup, use dentures, and go

**Table 2. Results of the OFI-8 questionnaire.**

| | Total (n = 251) | Men (n = 128) | Women (n = 123) | p |
|---|---|---|---|---|
| OFI-8 score | 3.3±2.1 | 2.8±2.0 | 3.8±2.0 | <0.001 |
| OFI-8 score≧4, n(%) | 97(38.6) | 37(28.9) | 60(48.8) | 0.002 |
| 1) Do you have any difficulties eating tough foods compared 6 months ago? (Yes) | 46(18.3) | 17(13.3) | 29(23.6) | 0.049 |
| 2) Have you choked on your tea or soup recently? (Yes) | 75(29.9) | 28(21.9) | 47(38.2) | 0.006 |
| 3) Denture use (Yes) | 144(57.4) | 65(50.8) | 79(64.2) | 0.041 |
| 4) Do you often experience having a dry mouth? (Yes) | 76(30.3) | 36(28.1) | 40(32.5) | 0.493 |
| 5) Do you go out less frequently than you did last year? (Yes) | 99(39.4) | 41(32.0) | 58(47.2) | 0.020 |
| 6) Can you eat hard foods like squid jerky or pickled radish? (No) | 29(11.6) | 11(8.6) | 18(14.6) | 0.167 |
| 7) How many times do you brush your teeth per day? (<3 times/day) | 47(18.7) | 28(21.9) | 19(15.5) | 0.200 |
| 8) Have you visit dental clinic at least annually? (No) | 42(16.7) | 22(17.2) | 20(16.3) | 0.867 |

Data are expressed as mean ± SD.

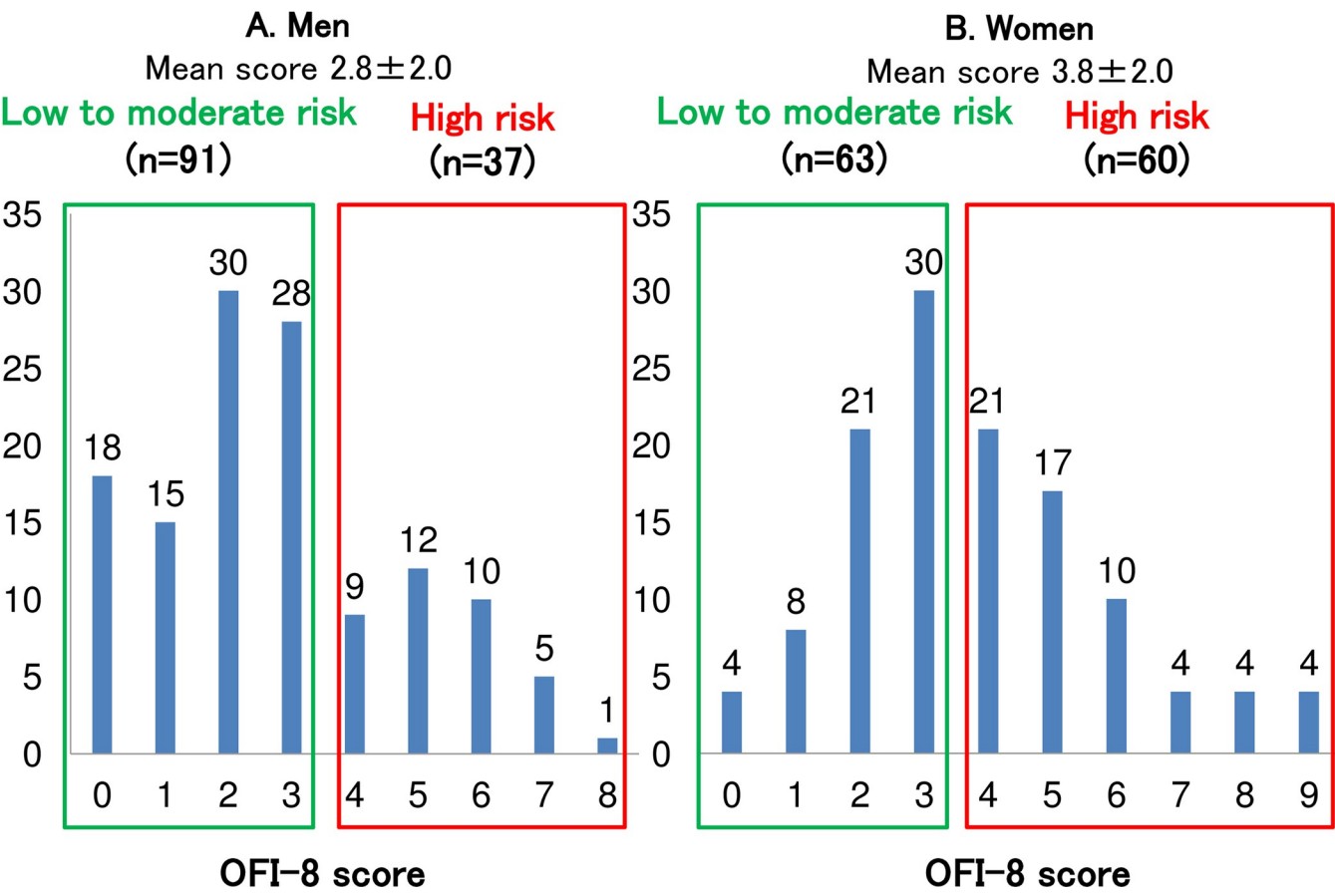

**Fig 1.** Distribution of OFI-8 scores in men (A) and women (B).

out less frequently than they did half a year ago. Men and women were almost equally likely to suffer from dry mouth and visit dental clinics. Men tended to have better oral function with respect to the other items; however, the differences were not significant.

The distribution of the OFI-8 scores is shown in Fig 1. The mean score was higher in women; therefore, the peaks of the distribution were also higher in women. The participants were divided into two groups according to their OFI score: low- to moderate-risk ($\leq$3 points) and high risk ($\geq$4 points) (Table 3). The average value of each parameter is presented. In men, the height, grip strength, Cr/CysC, eGFRcys/eGFRcre, red blood cell count, hemoglobin level, and albumin level were significantly lower in the high-risk group. In women, grip strength, Cr/CysC, and eGFRcys/eGFRcre were significantly lower in the high-risk group. Therefore, a significant difference was observed in grip strength, Cr/CysC and eGFRcys/eGFRcre between men and women. According to the ROC analysis, although the AUC was lower, Cr/CysC and eGFRcys/eGFRcre were significantly associated with a high risk of OFI-8 $\geq$ 4 points in both men and women (Fig 2).

## Discussion

Oral frailty is defined as a mild decline in oral functions during the early and reversible stages of frailty. Many community-dwelling older people have reduced oral function or oral hypo-function, which is significantly associated with frailty and aging. Appropriate evaluation of

**Table 3. Characteristics of participants with and without OFI-8 scores of ≥4.**

| | Men | | | Women | | |
|---|---|---|---|---|---|---|
| | OFI-8 score≦3 (n = 91) | OFI-8 score≧4 (n = 37) | p | OFI-8 score≦3 (n = 63) | OFI-8 score≧4 (n = 60) | p |
| Age (years) | 76.4±7.4 | 79.0±6.8 | 0.064 | 77.7±6.1 | 79.1±5.2 | 0.173 |
| Height (cm) | 165.8±6.9 | 162.5±7.3 | **0.018** | 152.1±5.7 | 150.4±6.1 | 0.115 |
| Weight (kg) | 66.1±9.3 | 63.1±9.8 | 0.107 | 52.1±9.1 | 52.0±9.5 | 0.949 |
| Body mass index: BMI | 24.0±2.6 | 23.8±2.6 | 0.673 | 22.6±4.1 | 23.0±3.8 | 0.574 |
| Grip strength (kg) | 30.7±7.6 | 27.0±8.5 | **0.018** | 17.7±4.5 | 15.3±4.2 | **0.002** |
| Cr (mg/dL) | 1.06±0.33 | 1.07±0.36 | 0.824 | 0.82±0.20 | 0.85±0.41 | 0.674 |
| CysC (g/dL) | 1.23±0.38 | 1.33±0.40 | 0.182 | 1.20±0.35 | 1.31±0.53 | 0.154 |
| Cr/CysC | 0.87±0.15 | 0.81±0.15 | **0.042** | 0.70±0.13 | 0.65±0.13 | **0.022** |
| eGFRcre (mL/min/1.73 m$^2$) | 57.4±15.3 | 57.0±18.7 | 0.910 | 54.3±13.8 | 56.3±17.4 | 0.471 |
| eGFRcys (mL/min/1.73 m$^2$) | 59.3±18.0 | 53.3±16.2 | 0.078 | 55.7±17.9 | 52.0±18.6 | 0.260 |
| eGFRcys/eGFRcre | 1.04±0.21 | 0.95±0.20 | **0.031** | 1.03±0.22 | 0.93±0.21 | **0.012** |
| Red blood cell (×10$^4$/μL) | 443±50 | 418±53 | **0.015** | 410±48 | 409±52 | 0.882 |
| Hemoglobin (g/dL) | 13.9±1.3 | 13.4±1.4 | **0.036** | 12.7±1.3 | 12.6±1.5 | 0.851 |
| Hematocrit (%) | 41.3±4.1 | 40.0±4.1 | 0.105 | 38.3±3.8 | 38.2±4.4 | 0.861 |
| Total protein (g/dL) | 7.1±0.4 | 7.1±0.4 | 0.904 | 7.1±0.4 | 7.2±0.5 | 0.870 |
| Albumin (g/dL) | 4.2±0.3 | 4.0±0.3 | **<0.001** | 4.2±0.2 | 4.1±0.3 | 0.271 |
| Hypertension, n(%) | 80(87.9) | 36(97.3) | 0.178 | 58(92.1) | 51(85.0) | 0.264 |
| Diabetes, n(%) | 21(23.1) | 8(21.6) | 1.000 | 8(12.7) | 13(21.7) | 0.233 |
| Dyslipidemia, n(%) | 41(45.1) | 23(62.2) | 0.118 | 41(65.1) | 36(60.0) | 0.581 |
| Osteoporosis, n(%) | 3(3.3) | 0(0.0) | 0.556 | 10(15.9) | 14(23.3) | 0.365 |
| Malignant neoplasm, n(%) | 3(3.3) | 2(5.4) | 0.626 | 1(1.6) | 2(3.3) | 0.613 |
| Cardiovascular disease, n(%) | 20(22.0) | 14(37.8) | 0.079 | 14(22.2) | 14(23.3) | 1.000 |
| Cerebrovascular disease, n(%) | 10(11.0) | 2(5.4) | 0.507 | 6(9.5) | 6(10.0) | 1.000 |

Data are expressed as mean ± SD.

oral function and effective intervention to suppress oral function deterioration may be effective in extending the healthy life expectancy of older people [29].

Frailty, in contrast, is considered a state of increased vulnerability to disease onset and physical dysfunction due to a decline in several functions associated with aging. Sarcopenia, a state of reduced muscle mass, is a typical physical frailty phenotype. Cystatin C is a strong predictor of organ-specific outcomes such as cardiovascular events, life outcomes [30], and renal outcomes [31]. One advantage of CysC over creatinine, which is also a marker of renal function, is that cystatin C is less affected by muscle mass.

As muscle mass decreases, Cr decreases relative to CysC, which is less affected by muscle mass. Therefore, cystatin C-related indices, such as Cr/CysC and eGFRcys/eGFRcre, decrease. Therefore, cystatin C-related indices have been demonstrated to be lower in physically frail individuals. In this study, we showed that cystatin C-related indices are also lower in high-risk states of oral frailty, a phenotype of physical frailty.

Frailty can be considered primary/pre-clinical when not directly associated with a specific disease and without significant functional disability. Therefore, frailty phenotypes (for example, oral frailty) could better define primary frailty and potentially be applied to a pre-clinical context, tailoring specific treatments and/or prevention strategies. Early intervention in frail older adults is important for improving clinical outcomes and reducing medical care. Many efforts have been made to develop subjective and objective tools for the early recognition of frailty and measuring its severity [32–34].

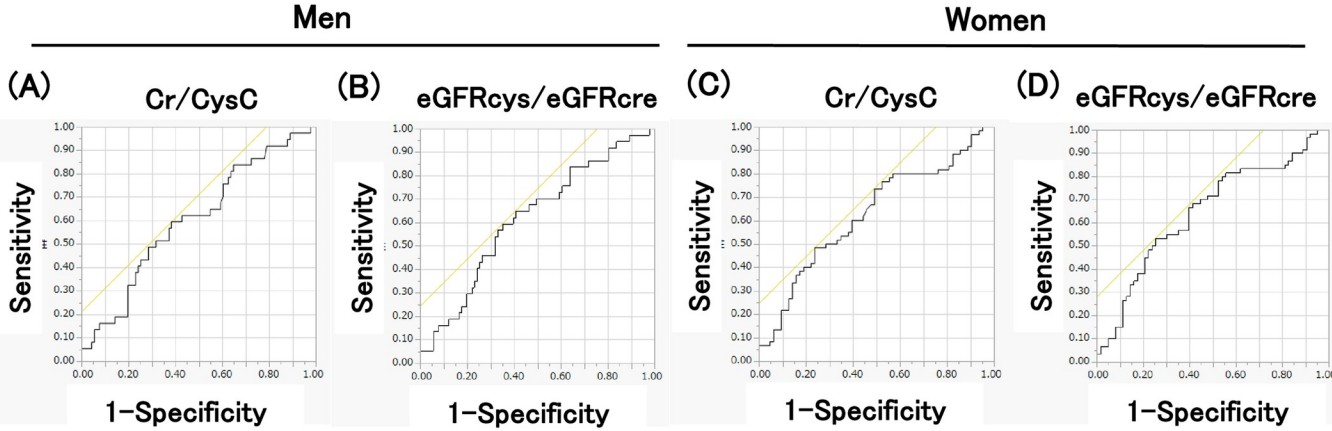

**Fig 2.** ROC and AUC of Cr/CysC (A) and eGFRcys/eGFRcre (B) for OFI-8≥4 in men. ROC and AUC of Cr/CysC (C) and eGFRcys/eGFRcre (D) for OFI-8≥4 in women. ROC, receiver operating characteristic curves. AUC, area under the curve.

Tanaka et al. reported that oral frailty is a significant risk factor for physical frailty, sarcopenia, nursing care needs, and total mortality and that the prognosis for life is also poor [3]. In 2018, the Japanese Society of Gerontology established criteria for assessing oral hypofunction using seven oral functions. The Japan Dental Association considers oral hypofunction to be a preliminary step in oral frailty, and the seven items listed here are used for objectively evaluating 1) oral hygiene, 2) oral dryness, 3) occlusal force, 4) tongue-lip motor function, 5) tongue pressure, 6) masticatory function, and 7) swallowing function, all of which require specialized dental equipment [4]. Therefore, an objective evaluation at the stage of oral frailty is difficult.

Therefore, simple and easy questionnaires, such as the Oral frailty index-8 (OFI-8), are very useful for early detection and treatment of oral frailty. Tanaka et al. have developed an oral frailty checklist using the OFI-8, an 8-item questionnaire [5]. Any difficulties eating tough foods compared to 6 months ago indicate a decline in chewing ability. Swallowing ability is indicated by difficulty in swallowing tea and soup; dry mouth, by xerostomia; going out less frequently, by decreased social participation; and the ability to chew foods as hard as squid jerky or pickled radish, by decreased masticatory ability. Tooth brushing and dental check-ups indicate oral hygiene-related behaviors.

In addition to the OFI-8, other questionnaires to assess oral frailty have been developed [35, 36]. Among them, the OFI-8 can be performed quickly and easily in a general medicine outpatient setting, and it does not require specialized skills, making it useful in screening for oral hypofunction. The OFI-8 has been proven to reflect oral function assessed objectively by oral examinations conducted by trained dental staff. The use of OFI-8 might support dental care practitioners in efficiently cooperating with communities and primary care providers, and

consequently help prolong healthy life expectancy and improve overall health by benefiting nutrition. Previous studies on the OFI-8 have focused on its use in epidemiological studies; however, this study shows that the OFI-8 can be easily used in a general internal medicine practice setting for assessing the risk of oral frailty.

Previous reports have shown that oral frailty is more common in women than in men; the overall prevalence of oral hypofunction has been reported to be 50.5%, with 40.3% in men and 54.9% in women. This result shows a trend similar to that observed for sarcopenia. The prevalence of sarcopenia was 18.6% overall, 9.7% in men, and 22.5% in women [37]. In another study, the prevalence of oral frailty was 35.8% in men, and 64.2% in women [38]. Additionally, the prevalence of systemic frailty has been reported to be higher in women than in men [39, 40]. These reports indicate that oral frailty, as well as systemic frailty, is more common in women than in men. The higher prevalence of the frailty in women may be because women have lower muscle mass and muscle strength than men. In the present study, participants at high risk of oral frailty (OFI-8≥4) were more likely to be women. This result is consistent with the previous reports described above.

In the present study, we divided the patients into two groups: low- to moderate-risk (up to 3 points) and high-risk (above 4 points) groups and examined whether there was a difference in the clinical indices. A score of ≥4 points indicates the necessity of a dental checkup, as older adults with these scores are at high risk of new-onset oral frailty and new long-term care needs certification.

In this study, grip strength and Cr/CysC and eGFRcys/eGFRcre were significantly lower in the high-risk oral frailty group (OFI-8≥4) in both men and women. This suggests that among the blood test indices, cystatin C-related indices such as Cr/CysC and eGFRcys/eGFRcre are the most versatile index related to oral frailty. In contrast, in men, RBC and Hb levels were significantly lower in the high-risk oral frailty group (OFI-8≥4). These results suggest that anemia may be significantly related to oral frailty in men as well. According to our previous reports, indices of anemia (RBC, hemoglobin, and hematocrit) exhibited a mild positive correlation with skeletal muscle mass index [41]. These results support a previous report, according to which anemia was associated with lower muscle strength and physical performance [42].

In men, albumin levels were also significantly lower in the high-risk oral frailty group (OFI-8≥4). Albumin level is an important indicator of nutritional status. Many reports have been published on the relationship between oral frailty and malnutrition. Older adults with multiple oral health problems have an increased risk of deterioration of nutritional status [34]. Malnutrition is associated with increased risks of frailty, sarcopenia, morbidity, and mortality [43–46]. Oral hypofunction is significantly and independently associated with protein intake in both men and women [47]. Community-dwelling older adults with oral frailty had an increased risk of deteriorating nutritional status, as evaluated using the Mini Nutritional Assessment Short Form (MNA-SF) [38]. Nomura et al. reported that oral health evaluated using the OFI-8 correlated with nutritional intake. The frequency of the items on the OFI-8 was similar to that in previous reports [48].

Grip strength was significantly lower in both men and women with OFI-8≥4. Base muscle mass and strength are lower in women than in men, and oral frailty may lead to muscle weakness even before progression of anemia and malnutrition in women.

There exists certain confusion and overlap between the concepts of physical and oral frailty. In addition, there exists an overlap among the diagnostic indicators. The OFI-8 index includes the question, "Do you have any difficulties eating tough foods compared to 6 months ago?", "Have you choked on your tea or soup recently?" and "Do you often experience having a dry mouth?" The same questions exist in the Kihon checklist, which was designed to screen for systemic frailty [49]. It is supposed that oral frailty, which is handled by dentistry, and physical

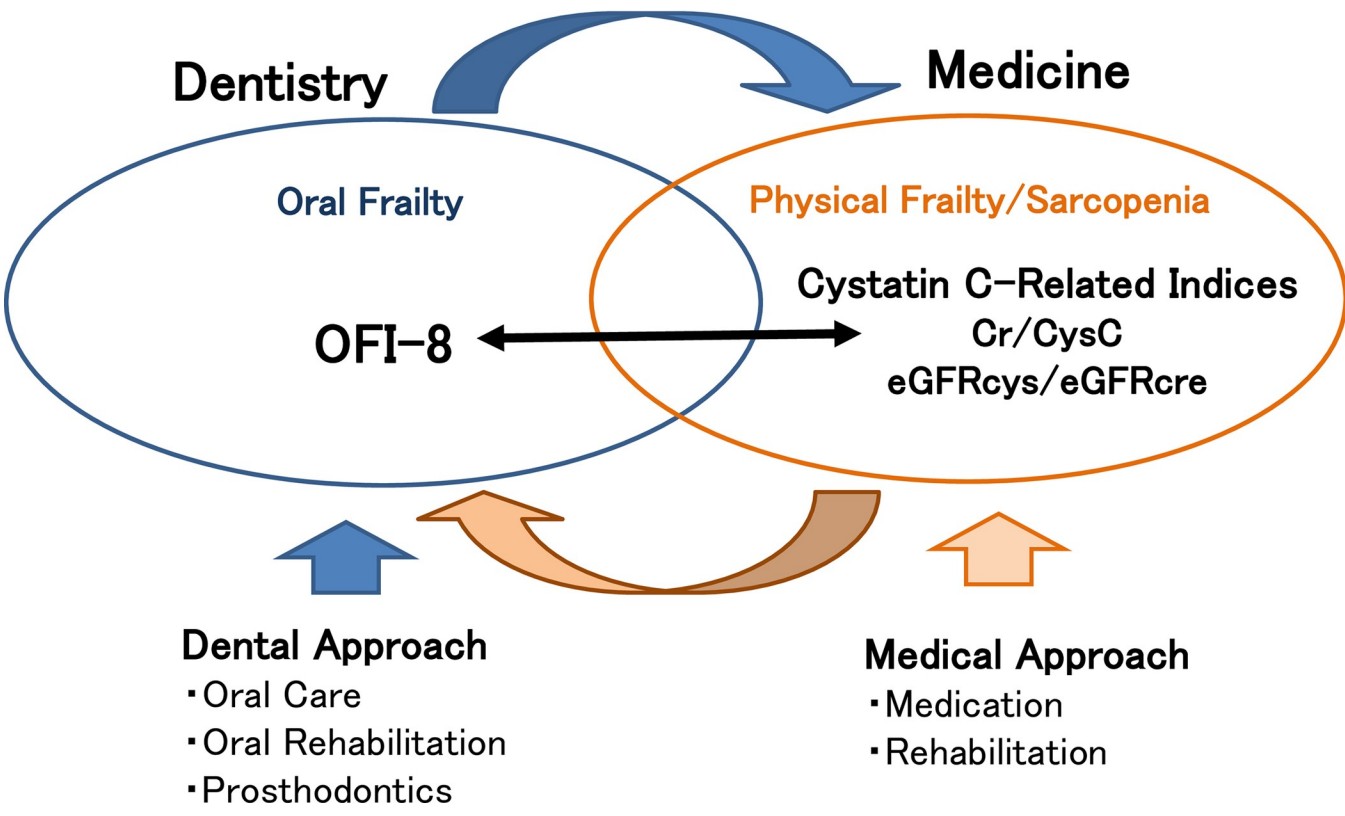

**Fig 3. Image of OFI-8 and cystatin C-related indices in medical-dental cooperation.**

frailty/sarcopenia, which is handled by medicine, overlap in some areas and form a vicious cycle that influences each other. We suppose that this vicious cycle can be inhibited by using dental approaches such as oral care, oral rehabilitation, and prosthetics for oral frailty; medical approaches such as medication and physical rehabilitation for physical frailty and sarcopenia; and a combination of both approaches (Fig 3). Thus, we would like to propose a bidirectional approach from both medical and dental perspectives.

For simple screening, the OFI-8 can be used for oral frailty and cystatin C-related indices such as Cr/CysC and eGFRcys/eGFRcre for physical frailty/sarcopenia; the present study shows that they are significantly associated with each other.

Recently, Kugimiya et al. reported that multiple aspects of oral functions including occlusal force, tongue-lip motor function, tongue pressure, masticatory function, and swallowing function were low among community-dwelling older adults with sarcopenia [50]. The group with a high OFI score and at high risk for oral frailty was found to have low grip strength, low muscle strength, and muscle mass, as reflected by the low values of cystatin C-related indices in the high-risk oral frailty group.

Low cystatin C-related indices suggest low skeletal muscle mass and a high risk of oral frailty. This finding suggests that low cystatin C-related indices are associated with low skeletal muscle mass and a high risk of oral frailty. If cystatin C-related indices assessed in the medical practice setting are low, it is recommended that oral frailty risk be assessed using the OFI-8 questionnaire. Moreover, if necessary, a precise oral function evaluation by a dentist should be recommended. The present study also demonstrated that hemoglobin and albumin levels were

low in the group identified as having a high risk of oral frailty by the OFI-8 index. These patients may have potential systemic diseases such as anemia and malnutrition.

This study reported an association between cystatin C-related indices and OFI-8 scores. A low cystatin C-related index is associated with a high risk of sarcopenia and oral frailty from a medical perspective, whereas a high OFI-8 score recommends a precise evaluation of oral functions from a dental perspective. Furthermore, this finding indicates the presence of sarcopenia, systemic frailty, or systemic diseases.

## Limitations

Our study has limitations that must be considered. First, this was a cross-sectional study, and we could not determine any cause-and-effect relationships. A follow-up prospective study is needed for assessing the causal associations between OFI-8 and cystatin C-related indices.

Because this study did not objectively assess oral function using the seven items specified by the Japan Dental Association, the exact oral function of those judged to be at high risk with an OFI-8 score $\geq 4$ is unknown. This is another limitation of this study. However, the purpose of this study was to determine whether an association exists between oral function assessed by a questionnaire and objective clinical indicators related to cystatin C, without the use of professional dental skills and equipment. Therefore, we do not believe that evaluation of oral function using specialized instruments and skills is necessary.

Next, there is room for improvement in the OFI-8. Nomura et al. reported that the following items of OFI-8 had low discrimination ability: "Brushing teeth at least twice a day," "Regular attendance of dental clinic," and "Using dentures." Therefore, they concluded that the OFI-8 scoring system needs to be improved [51].

Finally, we recruited only a small population sample from two medical institutes. This limits the reliability and validity of these tests.

## Conclusion

In conclusion, cystatin C-related indices were significantly lower in the group at high risk for oral frailty, identified using the OFI-8. Conventionally, cystatin C-related indices have been considered effective in screening systemic frailty and sarcopenia, whereas the OFI-8 index is considered effective in screening oral frailty. The present study demonstrated a relationship between the indices considered effective for screening both systemic and oral frailty, which have conventionally been evaluated separately from medical and dental perspectives. It is important to promote medical-dental collaboration by approaching systemic and oral frailty from both medical and dental perspectives.

## Acknowledgments

We would like to thank all the medical staff of Osaka Dental University and the National Cerebral and Cardiovascular Center who supported this study. We would like to thank Editage (www.editage.com) for English language editing.

## Author Contributions

**Conceptualization:** Keita Yamasaki, Masaharu Motone.

**Data curation:** Hiroshi Kusunoki, Kazumi Ekawa, Keita Yamasaki, Masaharu Motone, Fumiki Yoshihara, Hideo Shimizu.

**Formal analysis:** Hiroshi Kusunoki, Kazumi Ekawa, Nozomi Kato.

**Investigation:** Hiroshi Kusunoki, Kazumi Ekawa, Nozomi Kato, Ken Shinmura, Fumiki Yoshihara.

**Methodology:** Ken Shinmura.

**Supervision:** Hideo Shimizu.

**Writing – original draft:** Hiroshi Kusunoki.

**Writing – review & editing:** Hideo Shimizu.

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
