## [Decision Letter · Decision Letter 0]

25 Jan 2023

PONE-D-23-00128Association between Oral Frailty and Cystatin C-Related Indices - A Questionnaire (OFI-8) Study in General Internal Medicine Practice.PLOS ONE

Dear Dr. Kusunoki,

Thank you for submitting your manuscript to PLOS ONE. After careful consideration, we feel that it has merit but does not fully meet PLOS ONE’s publication criteria as it currently stands. Therefore, we invite you to submit a revised version of the manuscript that addresses the points raised during the review process.

This paper in the present form does not reach to an enough level for acceptance in Hypertension Research.  Major revisions are needed according to the Reviewers' comments.  AE did not find the second reviewer in this manuscript. However, the first reviewer showed the significant detailed suggestions. To avoid the delay of the fate, AE decides the fate from the first reviewer. Thus, see the comments carefully and respond them appropriately.

We look forward to receiving your revised manuscript.

Kind regards,

Masaki Mogi

Academic Editor

PLOS ONE

Reviewers' comments:

Reviewer's Responses to Questions

**Comments to the Author**

1. Is the manuscript technically sound, and do the data support the conclusions?

Reviewer #1: Yes

2. Has the statistical analysis been performed appropriately and rigorously? 

Reviewer #1: Yes

3. Have the authors made all data underlying the findings in their manuscript fully available?

Reviewer #1: Yes

4. Is the manuscript presented in an intelligible fashion and written in standard English?

Reviewer #1: Yes

5. Review Comments to the Author

Reviewer #1: In the present Manuscript, the Authors conducted a cross-sectional study on 251 patients (128 men and 123 women) aged ≥65 years admitted to the Osaka Dental University and the National Cerebral and Cardiovascular Center, between April 2022 and December 2022, to evaluate the association between cystatin C-related indices and oral frailty. Oral health is a neglected but important area in aging research and any contribution in this regard could be very important to highlight the potential association between oral health problems and health-related outcomes. However, the paper needs significant revision including:

Abstract

1. “Age-related deterioration in oral function (oral frailty)…”.

Right from the abstract it would be more appropriate to provide the correct definition of oral frailty, a frailty phenotype that has been recently suggested as a novel construct defined as a decrease in oral function guided by a cluster of impairments (i.e., tooth loss, periodontal disease, inadequate dental prostheses, difficulty in chewing, age-related changes in swallowing, etc.) that worsen oral daily practice functions with a coexisting decline in cognitive and physical functions (please see: Lancet Healthy Longev. 2021 Aug;2(8):e507-e520; J Gerontol A Biol Sci Med Sci. 2018 Nov 10;73(12):1661-1667).

2. Please include in the "methods" subsection of the abstract the median age of the patients and the design of the study.

3. I would replace "grip power" with grip “strength”.

4. I would recommend that the conclusions be modified based on the suggestions provided in the later sections of this review (please see point 8…)

Introduction

5. “Oral frailty, a minor deterioration of oral functions…”.

Again, oral frailty can be defined as a diminished oral function with a co-existing decline in cognitive and physical functions. The importance of this phenotype is linked to the potential reversibility of all oral health deficits and its relevant role as a risk factor for Alzheimer’s disease, other neurodegenerative conditions, and adverse major health-related outcomes (please see: Geroscience. 2022 Oct 15. doi: 10.1007/s11357-022-00663-8). Talking about potential reversibility, therefore, does not authorize defining oral frailty as a minor deterioration of oral functions.

6. “… is, therefore, a preliminary stage of physical frailty and sarcopenia”.

Oral frailty is certainly associated with physical frailty and sarcopenia but does not represent their preliminary stage. In the cited paper, Tanaka et al., right from the title, talk about oral frailty as a risk factor for physical frailty. The most widely used operational definition of frailty, the biological or syndromic construct of Fried and colleagues, describes a phenotype. This original phenotype is often referred to as physical frailty, to distinguish it from similarly constructed frailty syndromes for cognitive frailty, social frailty, and organ-specific frailties, as, in this case, oral frailty.

7. “By approaching and evaluating oral frailty from a medical perspective…”.

This concept was well expressed in a recent Editorial which underlined how frailty can be considered primary/pre-clinical when not directly associated with a specific disease and without a significant functional disability. Therefore, frailty phenotypes (e.g. oral frailty) could be able to better define primary frailty and may potentially be applied in a pre-clinical context, tailoring specific treatments and/or prevention strategies. In contrast, frailty may be secondary/clinical, a condition better defined using the deficit accumulation frailty model, if associated with accumulating multimorbidity, i.e., dementia, cerebral or cardiovascular diseases, and/or functional disability (please see: Curr Top Med Chem. 2022 Jun 15).

8. “… if an index that reflects oral function can be established using blood test data…”.

It would be more appropriate to talk about the association between oral frailty and cystatin C-related indices, which is the topic of the study, instead of considering the above indices as a sign of the concomitant presence of oral frailty, because the serum level of cystatin C is also a stronger predictor of other organ-specific outcomes, such as the renal outcome and the risk of cardiovascular events.

Methods

9. There is no information on who carried out the oral assessments. How many people? Were they doctors, dentists, or dental hygienists? Were the inter- and intra-observer agreements evaluated?

10. Can the Oral Frailty Index-8 be considered a checklist proposed by the Japan dental association since the paper by Tanaka et al. in which it was proposed is from 2021?

Discussion

11. “Minor deterioration of oral function in the elderly, such as increased choking, spills, and tongue deterioration…”.

Again, that is not the correct definition of oral frailty (please see points 1 and 5).

12. “Oral frailty is defined as a mild decline in oral function, and it occurs in the early and reversible stages of frailty”.

There is some confusion and overlap between the concepts of physical and oral frailty. More clarity is recommended (please see point 7).

13. “The combined use of these indices is expected to provide a multifaceted understanding and evaluation of both systemic and oral frailty”.

There is still an overlap between the two operational definitions of frailty i.e. the physical/biological construct of Fried and colleagues and the deficit accumulation model of Rockwood and Mitnitski. How can the above indices fit into the physical frailty model?

Furthermore, an association between cystatin C-related indices and oral frailty is not pathognomonic of the presence of oral frailty, since, as pointed out earlier, the same indices are also associated with other diseases and outcomes.

14. The limitations of the study sub-section should be implemented.

Conclusion

The conclusion is overstated. “The combined use of cystatin C-related indices and OFI-8 could facilitate screening for oral frailty at an early stage and prevent systemic sarcopenia/frailty.”

That is not correct. These are different assessments for two different outcomes that may or may not be associated.

6. PLOS authors have the option to publish the peer review history of their article (what does this mean?). If published, this will include your full peer review and any attached files.

Reviewer #1: **Yes: **Francesco Panza, MD, PhD

---

## [Author Response · Author response to Decision Letter 0]

9 Mar 2023

Response to reviewer #1

We would like to thank the reviewers for their constructive criticism and helpful suggestions, which helped us revise and improve our manuscript. We would also like to thank the reviewers for their valuable remarks.

In addition, the order of the review items has been slightly rearranged according to related items.

Reviewer #1: 

In the present Manuscript, the Authors conducted a cross-sectional study on 251 patients (128 men and 123 women) aged ≥65 years admitted to the Osaka Dental University and the National Cerebral and Cardiovascular Center, between April 2022 and December 2022, to evaluate the association between cystatin C-related indices and oral frailty. Oral health is a neglected but important area in aging research and any contribution in this regard could be very important to highlight the potential association between oral health problems and health-related outcomes. However, the paper needs significant revision including:

Abstract

1. “Age-related deterioration in oral function (oral frailty)…”.

Right from the abstract it would be more appropriate to provide the correct definition of oral frailty, a frailty phenotype that has been recently suggested as a novel construct defined as a decrease in oral function guided by a cluster of impairments (i.e., tooth loss, periodontal disease, inadequate dental prostheses, difficulty in chewing, age-related changes in swallowing, etc.) that worsen oral daily practice functions with a coexisting decline in cognitive and physical functions (please see: Lancet Healthy Longev. 2021 Aug;2(8):e507-e520; J Gerontol A Biol Sci Med Sci. 2018 Nov 10;73(12):1661-1667).

5. “Oral frailty, a minor deterioration of oral functions…”.

Again, oral frailty can be defined as a diminished oral function with a co-existing decline in cognitive and physical functions. The importance of this phenotype is linked to the potential reversibility of all oral health deficits and its relevant role as a risk factor for Alzheimer’s disease, other neurodegenerative conditions, and adverse major health-related outcomes (please see: Geroscience. 2022 Oct 15. doi: 10.1007/s11357-022-00663-8). Talking about potential reversibility, therefore, does not authorize defining oral frailty as a minor deterioration of oral functions.

6. “… is, therefore, a preliminary stage of physical frailty and sarcopenia”.

Oral frailty is certainly associated with physical frailty and sarcopenia but does not represent their preliminary stage. In the cited paper, Tanaka et al., right from the title, talk about oral frailty as a risk factor for physical frailty. The most widely used operational definition of frailty, the biological or syndromic construct of Fried and colleagues, describes a phenotype. This original phenotype is often referred to as physical frailty, to distinguish it from similarly constructed frailty syndromes for cognitive frailty, social frailty, and organ-specific frailties, as, in this case, oral frailty.

Poor oral health among the elderly is an important issue in general health, due to associations with the pathogenesis of frailty, which suggests a multidimensional geriatric syndrome.

11. “Minor deterioration of oral function in the elderly, such as increased choking, spills, and tongue deterioration…”.

Again, that is not the correct definition of oral frailty (please see points 1 and 5).

A. We have followed your suggestions and added the following text to the “Abstract” and “Introduction” sections. According to your suggestions, we have re-written the definition of oral frailty as follows:

Abstract　P2 L47-51

“Oral frailty is defined as an age-related gradual loss of oral functions, accompanied by a decline in cognitive and physical functions. It results in adverse health-related outcomes in older age, including mortality, physical frailty, functional disability, poor quality of life, and increased hospitalization and falls. 　Therefore, poor oral health among the elderly is an important health concern due to its association with the pathogenesis of systemic frailty, suggesting it to be a multidimensional geriatric syndrome. ”

Introduction P2 L84-93

“Oral health is an essential aspect of health, life satisfaction, QoL, and self-perception. Impairment of oral function is highly common in older adults, and aging has been reported to indirectly interact with several frailty domains through multiple pathways. An overt example of this relationship is age-related functional oral deterioration, characterized by poor dental hygiene, inadequate dental prostheses, and dietary deficiencies, resulting in a high risk of nutritional frailty(1). 　

Oral frailty is defined as an age-related gradual loss of oral function, accompanied by a decline in cognitive and physical functions. It results in major adverse health-related outcomes in older age, including mortality, physical frailty, functional disability, loss of quality of life, and reduced hospitalization and falls (2).

Poor oral health among the elderly is an important health concern due to its association with the pathogenesis of systemic frailty, suggesting it to be a multidimensional geriatric syndrome. Therefore, oral frailty may be a risk factor for systemic frailty (3).”

Discussion P5 

L199 

“Oral frailty is defined as a mild decline in oral functions during the early and reversible stages of frailty.”

L203-204

“Frailty, in contrast, is considered a state of increased vulnerability to disease onset and physical dysfunction due to a decline in several functions associated with aging.”

2. Please include in the "methods" subsection of the abstract the median age of the patients and the design of the study.

A. The mean and median ages are listed in the "Abstract" and "Materials and Methods" sections. We have mentioned the study design in the "Materials and Methods" section that this study was a cross-sectional study. Please check. (P4, L124-127)

“This was a cross-sectional study. All patients aged ≥ 65 years enrolled in this study were Japanese and provided informed consent to participate. We enrolled 251 patients with a mean age of 77.7±6.6 years and a median age of 77 years (128 men: mean age, 77.1±7.3 years; median age, 77 years and 123 women: mean age, 78.4±5.7 years; median age, 78 years)”

3. I would replace "grip power" with grip “strength”.

A. The term "grip power" has been replaced with "grip strength" throughout the manuscript Please check.

4. I would recommend that the conclusions be modified based on the suggestions provided in the later sections of this review (please see point 8…)

8. “… if an index that reflects oral function can be established using blood test data…”.

It would be more appropriate to talk about the association between oral frailty and cystatin C-related indices, which is the topic of the study, instead of considering the above indices as a sign of the concomitant presence of oral frailty, because the serum level of cystatin C is also a stronger predictor of other organ-specific outcomes, such as the renal outcome and the risk of cardiovascular events.

13. “The combined use of these indices is expected to provide a multifaceted understanding and evaluation of both systemic and oral frailty”.

There is still an overlap between the two operational definitions of frailty i.e. the physical/biological construct of Fried and colleagues and the deficit accumulation model of Rockwood and Mitnitski. How can the above indices fit into the physical frailty model?

Furthermore, an association between cystatin C-related indices and oral frailty is not pathognomonic of the presence of oral frailty, since, as pointed out earlier, the same indices are also associated with other diseases and outcomes.

A. In light of your comments, we have added the following text: Please check it. P5 L203-P6 L211

“Frailty, in contrast, is considered a state of increased vulnerability to disease onset and physical dysfunction due to a decline in several functions associated with aging. Sarcopenia, a state of reduced muscle mass, is a typical physical frailty phenotype. Cystatin C is a strong predictor of organ-specific outcomes such as cardiovascular events, life outcomes (30), and renal outcomes (31). One advantage of CysC over creatinine, which is also a marker of renal function, is that Cystatin C is less affected by muscle mass.

As muscle mass decreases, Cr decreases relative to CysC, which is less affected by muscle mass. Therefore, cystatin C-related indices, such as Cr/CysC and eGFRcys/eGFRcre, decrease. Therefore, cystatin C-related indices have been demonstrated to be lower in physically frail individuals. In this study, we showed that cystatin C-related indices are also lower in high-risk states of oral frailty, a phenotype of physical frailty.”

7. “By approaching and evaluating oral frailty from a medical perspective…”.

This concept was well expressed in a recent Editorial which underlined how frailty can be considered primary/pre-clinical when not directly associated with a specific disease and without a significant functional disability. Therefore, frailty phenotypes (e.g. oral frailty) could be able to better define primary frailty and may potentially be applied in a pre-clinical context, tailoring specific treatments and/or prevention strategies. In contrast, frailty may be secondary/clinical, a condition better defined using the deficit accumulation frailty model, if associated with accumulating multimorbidity, i.e., dementia, cerebral or cardiovascular diseases, and/or functional disability (please see: Curr Top Med Chem. 2022 Jun 15).

12. “Oral frailty is defined as a mild decline in oral function, and it occurs in the early and reversible stages of frailty”.　There is some confusion and overlap between the concepts of physical and oral frailty. More clarity is recommended (please see point 7).

A. In light of your comments, we have added the following text: Please check it.

P6 L212-214

“Frailty can be considered primary/pre-clinical when not directly associated with a specific disease and without significant functional disability. Therefore, frailty phenotypes (for example, oral frailty) could better define primary frailty and potentially be applied to a pre-clinical context, tailoring specific treatments and/or prevention strategies.”

P7 L271-275

“There exists certain confusion and overlap between the concepts of physical and oral frailty. In addition, there exists an overlap among the diagnostic indicators. The OFI-8 index includes the question, "Do you have any difficulties eating tough foods compared to 6 months ago?", “Have you choked on your tea or soup recently?” and “Do you often experience having a dry mouth?” The same questions exist in the Kihon checklist, which was designed to screen for systemic frailty (49).”

P7 L279-280

“Thus, we would like to propose a bidirectional approach from both medical and dental perspectives.”

P7 L284- P8 L299

“Recently, Kugimiya et al. reported that multiple aspects of oral functions including occlusal force, tongue-lip motor function, tongue pressure, masticatory function, and swallowing function were low among community-dwelling older adults with sarcopenia (50). The group with a high OFI score and at high risk for oral frailty was found to have low grip strength, low muscle strength, and muscle mass, as reflected by the low values of cystatin C-related indices in the high-risk oral frailty group.

Low cystatin C-related indices suggest low skeletal muscle mass and a high risk of oral frailty. This finding suggests that low cystatin C-related indices are associated with low skeletal muscle mass and a high risk of oral frailty. If cystatin C-related indices assessed in the medical practice setting are low, it is recommended that oral frailty risk be assessed using the OFI-8 questionnaire. Moreover, if necessary, a precise oral function evaluation by a dentist should be recommended. The present study also demonstrated that Hb and Alb levels were low in the group identified as having a high risk of oral frailty by the OFI-8 index. These patients may have potential systemic diseases such as anemia and malnutrition.

This study reported an association between cystatin C-related indices and OFI-8 scores. A low cystatin C-related index is associated with a high risk of sarcopenia and oral frailty from a medical perspective, whereas a high OFI-8 score recommends a precise evaluation of oral functions from a dental perspective. Furthermore, this finding indicates the presence of sarcopenia, systemic frailty, or systemic diseases.”

9. There is no information on who carried out the oral assessments. How many people? Were they doctors, dentists, or dental hygienists? Were the inter- and intra-observer agreements evaluated?

A. We have included the following text in the “Materials and Methods section. 　Inter- and intra-observer agreements were not assessed because we used a simple Yes or No questionnaire that asked the subjects to answer questions that would not have resulted in intra-observer differences. 

P4 L132-133

“Oral assessments using the questionnaire were performed by five internal medicine outpatient physicians.“

10. Can the Oral Frailty Index-8 be considered a checklist proposed by the Japan dental association since the paper by Tanaka et al. in which it was proposed is from 2021?

A. The Japan Dental Association proposed an index for diagnosing oral hypofunction that was published in 2018. The checklist in the Tanaka et al. paper was established in the Kashiwa study, an epidemiological study of the elderly at the University of Tokyo. Therefore, the checklist is not the one proposed by the Japan Dental Association. We have corrected the sentence as follows:

P4 L148-149

“The Oral Frailty Index-8 (OFI-8) was used in this study. The questionnaire consists of eight items and is widely used in Japan.”

14. The limitations of the study sub-section should be implemented.

A. A new section named “Limitation has been added. Please check.

Conclusion

The conclusion is overstated. “The combined use of cystatin C-related indices and OFI-8 could facilitate screening for oral frailty at an early stage and prevent systemic sarcopenia/frailty.”

That is not correct. These are different assessments for two different outcomes that may or may not be associated.

A. As you have pointed out, the use of the Cystatin C-related index in combination with the OFI-8 does not prevent systemic sarcopenia and frailty. Therefore, we conclude that our primary focus is on "the importance of approaching systemic frailty and oral frailty from both medical and dental perspectives and promoting medical-dental collaboration." 

P8 L319-323

“Conventionally, cystatin C-related indices have been considered effective in screening systemic frailty and sarcopenia, whereas the OFI-8 index is considered effective in screening oral frailty. The present study demonstrated a relationship between the indices considered effective for screening both systemic and oral frailty, which have conventionally been evaluated separately from medical and dental perspectives. It is important to promote medical-dental collaboration by approaching systemic and oral frailty from both medical and dental perspectives.”

---

## [Decision Letter · Decision Letter 1]

20 Mar 2023

Association between Oral Frailty and Cystatin C-Related Indices - A Questionnaire (OFI-8) Study in General Internal Medicine Practice.

PONE-D-23-00128R1

Dear Dr. Kusunoki,

We’re pleased to inform you that your manuscript has been judged scientifically suitable for publication and will be formally accepted for publication once it meets all outstanding technical requirements.

Kind regards,

Masaki Mogi

Academic Editor

PLOS ONE

Additional Editor Comments (optional):

The manuscript by Kusunoki et al. has been well-assessed by the Reviewer. It is ready to be accepted to PlosOne. No further comment.

Reviewers' comments:

Reviewer's Responses to Questions

**Comments to the Author**

1. If the authors have adequately addressed your comments raised in a previous round of review and you feel that this manuscript is now acceptable for publication, you may indicate that here to bypass the “Comments to the Author” section, enter your conflict of interest statement in the “Confidential to Editor” section, and submit your "Accept" recommendation.

Reviewer #1: All comments have been addressed

2. Is the manuscript technically sound, and do the data support the conclusions?

Reviewer #1: Yes

3. Has the statistical analysis been performed appropriately and rigorously? 

Reviewer #1: Yes

4. Have the authors made all data underlying the findings in their manuscript fully available?

Reviewer #1: Yes

5. Is the manuscript presented in an intelligible fashion and written in standard English?

Reviewer #1: Yes

6. Review Comments to the Author

Reviewer #1: All comments have been addressed. Thank you!

7. PLOS authors have the option to publish the peer review history of their article (what does this mean?). If published, this will include your full peer review and any attached files.

Reviewer #1: **Yes: **Francesco Panza, MD, PhD

---

## [Editor Report · Acceptance letter]

14 Apr 2023

PONE-D-23-00128R1 

Association between Oral Frailty and Cystatin C-Related Indices - A Questionnaire (OFI-8) Study in General Internal Medicine Practice. 

Dear Dr. Kusunoki:

I'm pleased to inform you that your manuscript has been deemed suitable for publication in PLOS ONE. Congratulations! Your manuscript is now with our production department. 

Kind regards, 

on behalf of

Dr. Masaki Mogi 

Academic Editor

PLOS ONE